# Identification of Auxin, Cytokinin, Transcription Factors, and Other Zygotic Embryogenesis-Related Genes in *Persea americana*: A Transcriptomic-Based Study [note 1]

**DOI:** 10.3390/plants14213288

**Published:** 2025-10-27

**Authors:** Ana O. Quintana-Escobar, Marcos David Couoh-Cauich, Brigitte Valeria Vargas-Morales, Martín Mata-Rosas, Eliel Ruíz-May, Víctor M. Loyola-Vargas

**Affiliations:** 1Unidad de Biología Integrativa, Centro de Investigación Científica de Yucatán, Calle 43, No. 130 x 32 y 34, Mérida CP 97205, Yucatán, Mexico; odetth@gmail.com (A.O.Q.-E.); md.couoh@gmail.com (M.D.C.-C.); vargasmbv@outlook.com (B.V.V.-M.); 2Red Manejo Biotecnológico de Recursos, Instituto de Ecología A.C., Carretera Antigua a Coatepec 351, Congregación El Haya, Xalapa CP 91073, Veracruz, Mexico; martin.mata@inecol.mx; 3Red de Estudios Moleculares Avanzados, Instituto de Ecología A. C, Carretera Antigua a Coatepec 351, Congregación El Haya, Xalapa CP 91073, Veracruz, Mexico; pelecas40@gmail.com

**Keywords:** avocado, embryo development, plant growth regulators, RNA-seq, seed

## Abstract

Zygotic embryogenesis is a key process in the development and propagation of avocado (*Persea americana*). Plant growth regulators, particularly auxins and cytokinins, play a crucial role in regulating this process. In this study, a transcriptomic analysis was performed to identify and characterize the expression of genes related to biosynthesis, transport, signaling, or response to auxins and cytokinins during different stages of embryonic development, as defined by the varying sizes of collected fruits. Additionally, several transcription factors and genes related to embryogenesis were analyzed. The results reveal dynamic patterns of gene expression that suggest a coordinated interaction between these PGRs in embryo formation and differentiation. This study provides key insights into the molecular mechanisms that regulate avocado zygotic embryogenesis, with potential applications in biotechnology and plant propagation.

## 1. Introduction

Avocado (*Persea americana*) is one of the most important fruits in the world. In recent decades, its consumption has increased among the population due to its nutritional value, as it is one of the fruits that naturally contains a high content of carbohydrates, proteins, and fatty acids, as well as different vitamins, minerals, and antioxidants. In recent years, the demand for this fruit has increased significantly in countries such as the United States, Europe, and China [1]. Among the central avocado-producing countries are Colombia, the Dominican Republic, Peru, Indonesia, Brazil, and Mexico; the latter is positioned as the leading producer with more than 28% of the world’s production, exceeding 2.9 million tons per year [2]. However, its cultivation is threatened by the social conflicts faced by producers [1].

On the other hand, social and cultural impacts, as well as the high demand for this fruit, have driven the development of studies focused on genetic and molecular improvement, as well as the genetic diversity of avocado [3]. For example, studies that include detection of restriction fragment length polymorphisms [4], analysis of randomly amplified polymorphic DNA [5], single nucleotide polymorphisms [6], simple sequence repeat markers derived from expressed sequence tags [7], as well as the use of RNA-Seq focused on flowering, development and fruit ripening has been carried out [8,9]. However, studies related to zygotic embryogenesis (ZE) in avocado are still limited, especially those using RNA-Seq approaches.

ZE represents a fundamental process in plant development. Here, the zygote is formed after double fertilization, marking the beginning of embryo development [10]. The study of zygotic embryo development has allowed a deeper understanding of the molecular, genetic, and physiological mechanisms that regulate plant development [11]. Most of these studies focus on the model plant *Arabidopsis* due to its sequenced genome and available genetic and molecular tools [12]. For example, the identification of molecular components related to zygote polarity [13], tissue differentiation, and embryo organization [14] active transcription in the zygote before the first cell division [15], as well as the interaction of plant growth regulators (PGRs) such as auxins and cytokinins, which play a crucial role in embryonic development [16]. It is well known that, during the early stages of embryogenesis, auxin establishes concentration gradients that determine apical-basal polarity, the orientation of cell divisions, and the establishment of the embryonic axis [17]. In parallel, cytokinins promote cell proliferation, participating in the formation of the apical meristem through the activation of *Arabidopsis Response Regulators*-type genes [18].

In recent years, sequencing technologies such as RNA-Seq have become established as fundamental tools for studying the transcriptome during different stages of embryonic development in model species such as *Arabidopsis*, maize, wheat, coffee, among others [15,19,20,21]. The study of ZE is of interest for various practices, including germplasm conservation, genetic improvement, identifying genes as potential markers, and its use in biotechnological techniques such as somatic embryogenesis (SE). Therefore, the implementation of RNA-Seq in non-model species such as avocado represents an opportunity to explore the molecular mechanisms that regulate embryogenic development. Thus, in the present study, we aimed to analyze the expression of genes associated with ZE in avocado using transcriptomic tools to provide key insights into the molecular mechanisms that govern this process.

## 2. Results and Discussion

A total of 24 libraries were constructed as a result of an eight-stage ZE paired-end sequencing with three replicates each. Since all of them were of high quality, it was decided to proceed directly to aligning with the reference genome, resulting in an average alignment rate of 91.9% (Appendix A). A total of 42,442 genes were identified from the avocado genome, of which 12,120 were not found to be expressed at any of the stages. These genes, whose expression values were zero in all samples, were excluded from subsequent analyses. A heatmap was created using the expression of the remaining 30,322 genes that were expressed in at least one stage (Figure 1A), revealing two distinct groups: the blue group representing genes with the lowest expression, and the magenta group corresponding to the most highly expressed genes. However, most of the genes have low expression. In general, a visual distinction can be made between stages 1 to 5 cm and 7 to 9 cm, since there are notable differences between the expression patterns.

The differential expression analysis was then performed (Figure 1B). The highest total number of DEGs was found in stage 7 cm with 6000 genes, of which 3970 were found upregulated and 2030 were downregulated. Followed by the last stages, 8 and 9 cm, with 5852 and 5654 DEGs, respectively; of which 3421 and 3410 were found to be upregulated, and 2431 and 2244 were downregulated. In the initial stages, the number of DEGs was significantly lower; for example, stage 2 cm had the lowest number, with 509 DEGs, of which 419 were upregulated and 90 were downregulated. This was expected, as it is the stage closest to 1 cm that was used for comparisons. Subsequently, stages 3 to 5 cm had 3587, 2379, and 1793 total DEGs, respectively. Of the former, 2239, 1154, and 863 were upregulated; while 1348, 1225, and 930 were downregulated. In general, we can conclude that, in most stages, there was a greater number of highly expressed genes, except for stages 4 and 5, where a slight difference was found, favoring the downregulated genes.

Only upregulated genes were selected to visualize gene interactions between the different stages using an UpsetR plot (Figure 1C). Notably, only 12 genes were shared among all samples. In contrast, the highest number of shared genes (1542) was found in stages 7–9 cm. The stage with the highest number of unique genes, i.e., not found in any other stage, is stage 7 cm with 912 genes, followed by stage 8 cm with 858, and stage 9 cm with 749.

A blast analysis was performed against *Arabidopsis* to assign functions to the genes. Homology was found for 25,993 genes, of which approximately 10% had null expression values at all stages. These genes, omitting those with zero value in all sizes, were selected for subsequent analysis. Additionally, the gene annotations and expression levels in TPM are presented in Appendix A.

We performed a GO enrichment analysis using the upregulated DEGs within each stage to gain insights into their potential biological meaning. At first glance, GO enrichment analysis of DEGs reveals genes involved in hormone-related stimulus, including gibberellic acid-mediated signaling pathway (GO:0009740), response to abscisic acid (GO:0009737), and ethylene-activated signaling pathway (GO:0009873). The participation of plant growth regulators in zygotic embryogenesis is well documented [17,22]. Moreover, DEGs at distinct stages show GO term enrichment for genes involved in stress responses such as cellular response to hypoxia (GO:0071456), response to fungus (GO:0009620), response to wounding (GO:0009611), response to salt stress (GO:0009651), response to cold (GO:0009409) and cold acclimation (GO:0009631); supporting biotic and abiotic stress influence during embryogenesis. Most enriched terms belonging to biological process (Figure 2A) associated with the 3 cm and 4 cm stages were predominantly photosynthesis-related processes such as photosynthesis (GO:0015979), photorespiration (GO:0009853), photosynthesis, light harvesting in photosystem I (GO:0009768), and photosynthesis, light harvesting (GO:0009765); which can suggest that the embryo is establishing its initial metabolic capacity and the establishment of energy capture systems. GO terms associated with the 7 cm, 8 cm, and 9 cm stages were cell division (GO:0051301), microtubule-based movement (GO:0007018), and regulation of DNA-templated transcription (GO:0006355) processes; possibly indicating an active gene expression regulation as the embryo establishes its developmental program and a cytoskeletal reorganization during embryo growth.

Similarly to the biological process enrichment, GO terms in the molecular function and cellular components categories, such as chlorophyll binding (GO:0016168) and chloroplast (GO:0009507), respectively, were also enriched. Nonetheless, photosynthesis during embryogenesis is an intriguing process that remains unclear and is rarely studied [23]. In a comparative proteomic study of *P. americana* seed tissues, proteins related to photosynthesis were detected as upregulated in the testa and cotyledons [24].

Among the GO terms belonging to molecular function (Figure 2B) enriched predominantly in the last stages (7–9 cm) were microtubule binding (GO:0008017), protein binding (GO:0005515), DNA-binding transcription factor activity (GO:0003700), DNA-binding (GO:0003677), and transcription cis-regulatory region binding (GO:0000976). These terms, alongside sequence-specific DNA binding (GO:0043565), were also found enriched at the 2 cm stage, which could suggest active transcriptional programming during early embryogenesis. Additionally, DEGs at these stages are predominantly associated with cellular components (Figure 2C), including plasmodesmata (GO:0009506), plant-type cell walls (GO:0009505), plasma membranes (GO:0005886), microtubules (GO:0005874), and nuclei (GO:0005634). These results highlight a close link to cellular division events during embryogenesis.

Regarding the cellular component (Figure 2C), an enrichment was found in ribosome-related terms. These terms, found in early or mid-stages, could be associated with protein-related molecular function terms, particularly in stages of active growth where greater protein synthesis is required for the development of the new embryo. Cellular components related to photosynthesis and plastid organelles were also found enriched, such as thylakoid (GO:0009579), chloroplast stroma (GO:0009570), and chloroplast (GO:0009507), reaffirming the importance of establishing the photosynthetic machinery. Terms indicating essential components for cell–cell communication, cell wall formation, and cytoskeleton organization in plants were also found, such as plasmodema (GO:0009506), plant-type cell wall (GO:0009505), plasma membrane (GO:0005886), and microtubule (GO:0005874).

We also performed a KEGG enrichment analysis of the upregulated DEGs to identify significantly enriched metabolic pathways. As shown in Figure 2D, KEGG analysis resulted in seventeen distinct enriched metabolic pathways. At the 2 cm stage, the top 5 included metabolic pathways (ath01100), biosynthesis of secondary metabolites (ath01110), plant hormone signal transduction (ath04075), MAPK signaling pathway-plant (ath04016), and phenylpropanoid biosynthesis (ath00940). The 3 cm and 4 cm stages share some pathways in common, such as ribosome (ath03010), photosynthesis-antenna proteins (ath00196), and oxidative phosphorylation (ath00190). The nucleotide excision repair (ath03420), ribosome (ath03010), porphyrin metabolism (ath00860), and fatty acid degradation (ath00071) were among the enriched pathways at the 5 cm stage. Moreover, 7 cm, 8 cm, and 9 cm stages shared pathway enrichments in biosynthesis of secondary metabolites (ath01110), metabolic pathways (ath01100), motor proteins (ath04814), and DNA replication (ath03030).

Genes involved in Aux and Ck metabolism, transcription factors, and others related to embryogenesis were identified among the DEGs.

Within the classification of Aux-related DEGs (Figure 3 and Appendix A), we were able to identify the *ATP-BINDING CASSETTE* (*ABC*), *AUXIN BINDING PROTEIN 1* (*ABP1*), *AUXIN RESPONSE FACTOR* (*ARF*), *GRETCHEN HAGEN 3* (*GH3*), *AUXIN/INDOLE-3-ACETIC ACID* or *INDOLE-3-ACETIC ACID INDUCIBLE* (*Aux/IAA, IAA*), *INDOLE-3-BUTYRIC ACID RESPONSE* (*IBR*), *LIKE AUXIN RESISTANT* (*LAX*), *PIN-FORMED* (*PIN*), *SMALL AUXIN UPREGULATED RNA* (*SAUR*), *TRYPTOPHAN AMINOTRANSFERASE RELATED* (*TAR*), and *YUCCA* (*YUC*) families.

There are several Aux receptors, among which *AUXIN BINDING PROTEIN 1* (*ABP1*) stands out. This protein has been found on the cell surface, having a high affinity for Aux [25]. We identified one *ABP1* gene that was only expressed at a depth of 3 cm. A mutation in this gene interrupts cell division and elongation to continue the development of the globular embryo [26]. Also, *ABP1* regulates early auxin-responsive genes such as *GH3* and *SAUR* [27], as well as fast phosphorylation of different proteins, including PIN2 [25].

According to the IAA perception, signaling is primarily mediated by *ARFs* and *IAAs*. Although these are widely studied gene families, the vast majority of studies reporting these analyses were conducted in somatic embryogenesis. In a survey carried out on somatic embryos of *P. americana* originated from zygotic embryos, they identified some *ARFs* that were also present in our system. They identified 20 *ARFs* in the genome-wide analysis [28]. We found a total of 17 *ARFs* in our transcriptome. We found that *ARF19a* and *ARF5* were consistently upregulated, starting at a fruit size of 7 cm. *ARF6a*, *ARF9*, *ARF6c-d*, and *ARF17* were specifically expressed at stages 3, 5, and 7 cm, respectively. *ARF5* is probably the most studied member of the family. This gene, also called *MONOPTEROS*, regulates multiple plant developmental processes, including zygotic embryogenesis. It controls the expression of several genes involved in embryo formation, and its absence results in deformed embryos [29]. We also identified a group of *ARFs* that remained downregulated throughout all stages, especially at later stages, which coincides with the previously mentioned study in avocado [28]. Additionally, no differentially expressed *ARFs* were identified in the 2 cm fruits. *Aux/IAAs* are auxin early response genes that repress the expression of *ARFs* by binding to them under low auxin concentration conditions [30]. Twelve *IAAs* were differentially expressed at every developmental stage except for the 5 cm size group. Most of them had the highest expression levels at the late stage (7–9 cm). It is proposed that diverse plant development processes are mediated by the paired interaction of *ARFs*-*Aux/IAAs* and their multiple combinations. The most studied combination is between *ARF5* and *IAA12*, with *IAA13* being its closest homolog. Therefore, *ARF5/IAA13* is an optimal pair for transcriptionally regulating embryogenesis [31]; we found that they were differentially expressed as fruit size increased in our model. Only *IAA4a* and *c* were downregulated at 3, 4, and 7 cm. The observed scenario where *ARFs* are repressed and *Aux/IAAs* are expressed in the later stages could suggest that this is a transitional stage in which the embryonic machinery is prepared before responding to Aux to activate the expression of responsive genes, which will subsequently be activated.

*PINs* are also necessary for Aux homeostasis. They carry out polar asymmetric Aux transport, achieving maximum or minimum levels for plant development processes such as embryogenesis, by establishing the embryonic axis [32]. Of the 10 *PINs* identified as DEGs, *PIN1*(*a*-*d*) and *PIN6* were most highly expressed in the late stage, starting at 7 cm; the latter gene showed the highest and almost sustained expression levels. In contrast, *PIN2a* was only found to be downregulated at sizes 4, 7, and 9 cm. No *PINs* were expressed at a size of 2 cm. *PIN1* is highly important in embryogenesis. Its role has been demonstrated with loss-of-function mutants in *Arabidopsis*, which resulted in deformed embryos with asymmetric cotyledons or lacking them [33]. In *Zea mays*, *PIN1* was localized to the central axis of mature embryos, in cells that later form vascular tissue [34]. It is worth mentioning that *PIN1* and *PIN2* have been found to act similarly in the same cells of *Arabidopsis*, suggesting that they may replace each other [35]. Although *PIN6* is the least studied member, it has been shown that it controls auxin homeostasis during embryogenesis, through efflux transport, identified in the vasculature and basal part of the mature embryo [36].

Belonging to the *ABC* family, we found 22 genes divided as follows: 5 from the B subfamily, 2 from the F subfamily, 12 from the G subfamily, and 3 from the I subfamily. Although the confirmed subfamilies involved in the transport of plant growth regulators are B and G, we decided to evaluate the remaining members. During the early stages of development, the *ABCG21*, *ABCG9a*, *ABCB15,* and *ABCB29* genes were expressed. At the intermediate stages, the genes that showed an increase in expression were *ABCG21*, *ABCG15*, *ABCI7*, *ABCI8*, and *ABCB1*, of which the last four were upregulated only at a size of 5 cm. The role of some members of the subfamily I as CK transporters has been suggested [37]. In the late stages, from 7 to 9 cm, *ABCG21*, *ABCG9a*, *ABCB4*, *ABCG16,* and *ABCB19* presented the maximum expression values. It is well known that the B subfamily, and some members of the G family, function as Aux influx and efflux transporters [29]. *ABCG9* was found to be expressed in the vascular system of the cotyledons of *A. thaliana* [38]. Also, *ABCG21* has been demonstrated to have a strong participation in IAA transport [39]. The best characterized Aux-transporting *ABCBs* in *Arabidopsis* are *ABCB1*, *ABCB4*, and *ABCB19* [40], which we identified in the intermediate and late stages of development, suggesting a late response to Aux mobilization for embryo development. In particular, *ABCB19* has been shown to intervene in the elongation of the hypocotyl of *A. thaliana* [41] and in the expansion of the cotyledons [42]. Furthermore, during the development of the ZE of *Paeonia ostii*, it was identified as one of the central nodes of the embryonic regulatory network [43]. Although the G subfamily does not appear to be directly related to the development of the zygotic embryo, it is involved in the mobilization of other cellular components (such as lipids, waxes, and suberin) towards the cell wall of various tissues, thus acting as a protective barrier against different types of stress in roots and seeds, such as desiccation and pathogens [44]. It is important to note a group of genes belonging to the G subfamily that remained negatively expressed throughout most stages, including *ABCG20*, *ABCG15*, *ABCG16*, *ABCG9b,* and *ABCG23*.

Additionally, we identified two copies of a member of the *AUX1/LAX* family, *LAX2*, both of which are primarily expressed in the late stages of development. This is a family of Aux influx carriers involved in the formation of the vasculature in zygotic embryos; in consequence, a mutation in this gene results in embryos with a discontinuous pattern in the vascular system of the cotyledons [45]. Specifically, *LAX2* is expressed in the embryo, participating in germination [46].

Furthermore, we found two *IBR* genes that participate in the β-oxidation of IBA to form IAA. Among the most characterized *IBRs*, *IBR1* and *IBR10* were found to be differentially expressed at the end of embryo development, which are known to be involved in cotyledon expansion and root elongation [47,48].

Nine genes belonging to the *GH3* family were identified. These genes are involved in the superpathway of IAA conjugate biosynthesis/inactivation (Figure 4). No expression levels were detected for most of these genes in the early stages. In contrast, towards the later stages, 7 to 9 cm, expression began for *GH3.1a-c*, *GH3.1e*, *GH3.11b*, and *GH3.6*, with the highest values at 8 cm. Little information is available on the participation of *GH3* during zygotic embryogenesis. However, in *P. ostii, GH3.11* was identified as actively expressed during the formation of the zygotic embryo [43]. The *GH3.1d* and *GH3.10* genes had negative values at these same stages. In the IAA inactivation pathway and conjugate biosynthesis, *GH3.1* intervenes in the conjugation of IAA with glutamic acid (Glu), aspartic acid (Asp), and phenylalanine (Phe); *GH3.6* participates in the conjugation with the same amino acids, in addition to glutamine (Gln) and valine (Val) (Figure 4). Similarly, in the somatic embryogenesis process of *Coffea canephora*, *GH3.1* and *GH3.6* were identified as participating during induction. Their high expression was related to high accumulation levels of conjugates such as IAA-Glu [49]. A similar pattern was found in *Tectona grandis*, where the expression levels of the *GH3* family increased during SE induction, correlating with the formation of IAA-Asp [50]. In *Arabidopsis*, it has been shown that GH3 mutants overaccumulate these two conjugates, thereby clarifying the role of these genes in auxin homeostasis [51]. The above suggests that for embryogenesis to occur, IAA levels within the cell must be perfectly regulated until homeostasis is achieved; one pathway involves the conjugation of free auxin with amino acids for either storage or degradation.

*SAUR* is the most prominent family of early auxin-responsive genes [52]. We identified 18 genes, in which a pattern was observed, with the majority of them expressed starting at a size of 7 cm. Of these, *SAUR72*, *SAUR12*, *SAUR52a*, and *SAUR42b* were the most highly expressed in the later stages of development. In contrast, *SAUR36a* and *SAUR71* remained repressed. Despite being a large and important family in various processes of plant development, its characterization and in-depth study have been limited, presumably due to the high redundancy among its members [52]. It is highly expressed during SE in *Gossypium hirsutum* [53] and *Dimocarpus longan* [54].

Aux biosynthesis during the ZE is mediated by the *YUC* and *TAA1/TAR* families. We identified one member from the *TAR* (*TRYPTOPHAN AMINOTRANSFERASE RELATED*) family highly expressed in the last two sizes. From the *YUC* family, we identified *YUC3*, *YUC4*, and *YUC10*, with two copies designated as *a* and *b*. The behavior was similar to that of the previously mentioned families, with the highest expression of *YUC3*, *YUC4*, and *YUC10a* observed from a size of 7 cm onward. Interestingly, copy *b* of *YUC10* was found to be repressed at these exact sizes. *YUC4* and *YUC10* are actively involved during SE and ZE of *Arabidopsis* [55]. Within the IAA biosynthesis pathway, *TAR2* catalyzes the conversion of tryptophan to Indole-3-pyruvate, which is subsequently converted to IAA by *YUC2*, *YUC4*, and *YUC10* (Figure 4). It has been found that the high expression of Aux biosynthesis genes is related to an increased concentration of IAA in SE [56].

Within the classification of CKs-related genes (Figure 5), those belonging to the *HISTIDINE KINASE* (*AHK*), *ARABIDOPSIS RESPONSE REGULATOR* (*ARR*), *CYTOKININ OXIDASE* (*CKX*), *CYTOKININ RESPONSE FACTOR* (*CRF*), *EQUILIBRATIVE NUCLEOSIDE TRANSPORTER* (*ENT*), *ISOPENTENYLTRANSFERASE* (*IPT*), and *PURINE PERMEASE* (*PUP*) families were detected.

Key genes, such as *AHK* and *ARR*, primarily drive CK signaling. Among the *AHKs*, members *AHK1*, *AHK2*, and *AHK5* (*a*, *b*) were found. Of these, *AHK1* was expressed at 3 cm and again at 7 cm and above, representing the highest expression levels for this family. *AHK5a* and *AHK2* were also found to be upregulated in the later stages, whereas *AHK5b* was only downregulated at the 7 cm stage. In *Arabidopsis,* its biological function focuses on the final development of the embryo and the growth of the stem and root [57]. *AHK2* is the general receptor for CKs, while others are organ-specific, such as *AHK4* [58]. For its part, the function of *AHK5* in the signaling system is not thoroughly studied, and its activity as a histidine kinase does not appear to be dependent on CKs [59]. *AHK1* and *CKI1* (*CYTOKININ-INDEPENDENT 1*) are structurally similar; they are histidine kinase receptors that participate in the two-component signaling system of CKs. *CKI1* was upregulated at 8 and 9 cm; however, it is not a direct receptor for CKs, but instead participates in a phosphorylation mechanism that initiates the signaling pathway. It has been suggested that they may have redundant functions with *AHKs* [60]. Likewise, this gene is involved in regulating cell expansion during vegetative growth [61]. Among the *ARRs*, we identified a total of 21 genes, including some copies. A peculiar behavior was observed in the members of this family, since practically all of them were found to express themselves positively. A group of *ARRs* was expressed during the 3 cm stage, and again from 7 cm onwards (*ARR1a*, *ARR2a, ARR22,* and *ARR24a-c*), while others were only expressed in the late stage (*ARR9a-b*/*d*, *ARR1b*, *ARR11c*, and *ARR10*). *ARR9c* had a unique behavior, as it was only observed to be expressed at the largest size, with the maximum value for this gene family. In contrast, *ARR9e* was repressed from 5 cm onwards. Within group B of *ARRs*, members *ARR1*, *ARR2*, *ARR10,* and *ARR12* regulate most genes induced by plant growth regulators and activate some transcription factors such as *WUS* [62]. During SE in *C. canephora*, three *ARR* group B (*ARF3*/*9*/*17*) were also found highly expressed [63].

*CRFs* are a type of transcription factor (TF) whose expression is mediated by *ARRs* and is involved in multiple plant developmental processes [64]. We identified *CRF2* and *CRF4a*/*b*. *CRF4a* and *CRF2* showed high expression starting at 4 cm size, while *CRF4b* was punctually repressed at 5 cm size. These genes, in turn, are related to the auxin pathway, particularly regulating its transport through *PINs* [65].

IPTs are key enzymes involved in CK biosynthesis. Two copies of *IPT1* and two of *IPT3* were identified. *IPT3a* was highly expressed at early and intermediate stages, and its expression subsequently increased at later stages. In contrast, *IPT3b* and *IPT1b* were found to be continuously downregulated from 7 cm onward. Similarly, in *Arabidopsis*, *IPT1* and *IPT3* were found to be downregulated by CKs [66]. In the CK biosynthesis pathway, specifically *trans*-zeatin, *IPT1* intervenes in the conversion of DMAPP to *N^6^*-(Δ^2^-isopentenyl)-adenosine 5′-triphosphate or diphosphate, to subsequently be transformed into *trans*-zeatin riboside triphosphate or diphosphate, respectively, by the action of the enzyme *CYP735A1* (*CYTOCHROME P450, FAMILY 735, SUBFAMILY A*), which was only expressed at the last size. In an alternative pathway, in addition to *IPT1*, *IPT3* is involved in converting DMAPP to *N^6^*-prenyladenosine 5′-phosphate, and subsequently to *trans*-zeatin riboside monophosphate (Figure 4). The *IPT3* expression pattern is localized in the root vasculature, specifically in the zone of highest differentiation [67].

*CKX* regulates CK levels within the cell by degrading them [60]. We identified two *CKX* genes: *CKX1*, with four copies, and *CKX5*, with three. Of these, *CKX5a* and *CKX1c* had size-specific expression at 7 cm; whereas *CKX5b* and *CKX1a* remained highly expressed at sizes 7–9 cm. *CKX1b* was found to be repressed at the mid-stage (4–5 cm) and subsequently activated from 7 cm onwards. *CKX5c* had sustained negative expression throughout the late development stage. *CKX1* and *CKX5* are involved in the irreversible degradation pathway of CKs (Figure 4). *CKX1* reduces the CK levels to a third of their initial content [68]. Also, double mutants of *CKX3* and *CKX5* increased CK content in reproductive tissue [69].

Also, we could find the first identified member in plants of the *ENT* family as a nucleoside transporter: *ENT1,* with three copies [70]. *ENT1c* was strongly activated at the late stage, while *ENT1a* was promptly repressed at a size of 7 cm. A different member, *ENT2*, in *Oryza sativa* facilitates the embryo’s uptake of nucleosides from the endosperm and, later, in plants, participates in long-distance transport [71].

*PUPs* are another type of nucleoside transporter for CKs. From this family, we identified *PUP1*, *PUP3* with three copies, *PUP5*, and *PUP11*. *PUP3a* and *PUP3b* behaved similarly, with negative expression at the 4 cm stage, becoming activated at 5 cm, and then becoming repressed again thereafter. In contrast, *PUP1* and *PUP11* were highly expressed from 7 cm onward. *PUP5* and *PUP3c* were expressed specifically at the 7 cm stage, both positively and negatively, respectively. The role of *PUP1* and its relationship with CKs was demonstrated with complementation assays in yeast [72]. However, although its direct participation during embryogenesis is not reported, in *O. sativa,* the low content of CK nucleosides in the shoot of plants with overexpression of *PUP1* was associated with the dwarf phenotype and low grain weight [73]. This could be because overexpression of *PUP1* impaired the transport of CKs from the root to the shoot [73]. In contrast, *PUP11* regulates seed development and grain filling by influencing the content of CKs, activating genes related to sugar transport and starch accumulation [74].

Last but not least, we identified the *CYTOKININ-RESPONSIVE GATA FACTOR 1* (*CGA1*) and *ADENOSINE KINASE 2* (*ADK2*). *CGA1* was upregulated at 7 cm. In *Arabidopsis* embryos, *CGA1* was found in cotyledons at intermediate stages of development: heart-torpedo; while in seedlings, it was found in all active growth zones and the vasculature, and it is also regulated by type B *ARRs*, specifically *ARR1* and *ARR12* [75]. *ADKs* are part of CKs’ homeostasis, as they participate in the interconversion of ribosides to nucleotides to regulate intracellular levels of CKs [76]. Here, *ADK2* had a constant positive expression from 3 cm onwards, except at 5 cm, where it was drastically downregulated. This last enzyme catalyzes the phosphorylation of adenosine to AMP in the adenine and adenosine salvage pathway (Figure 4). Little is known about its role in plant development [77]. Still, it was confirmed that Arabidopsis mutants lacking *ADK* resulted in embryo lethality [76], and that *ADK1* intervenes in root morphology [77].

A total of 1798 TFs were identified in the avocado genome, of which 512 were found to be upregulated in at least one stage. *bHLH* was the family with the highest number of members in the genome (157) (Appendix A). Some of those TFs are also found among the master regulators in embryogenesis (Figure 6). In the classification of TFs (not included in the other defined groups) we could identify large families such as *AGAMOUS/-like* (*AG/AGL*), *WUSCHEL-RELATED HOMEOBOX* (*WOX*), *WRKY*; as well as members of the *APETALA* superfamily (*AP2/ERF*, *APETALA2/ETHYLENE RESPONSE FACTOR*), such as *AP2/B3-like*, *ARIA* (*ARIA-INTERACTING DOUBLE AP2 DOMAIN PROTEIN*), *ERF*, *RAP2* (*RELATED TO AP2*) and *AP3* (*APETALA 3*).

The largest TF family identified was *WRKY* with 42 genes. Some of them were active only in the early and intermediate stages (*WRKY2*, *WRKY4*, and *WRKY48a*), while others were also positively expressed in the late stage (*WRKY28*, *WRKY31*, *WRKY40a*, *WRKY41a*, *WRKY42a-b*, and *WRKY51*). WRKY9, *WRKY7b*, *WRKY11*, *WRKY14*, *WRKY22*, *WRKY33a*, *WRKY43*, *WRKY44*, *WRKY65c-d*, *WRKY69* and *WRKY75a* had repressor activity at different sizes. The remaining TFs were found to be upregulated in the late stage. *WRKYs* are involved in multiple plant developmental processes, including dormancy, germination, flowering, and grain yield, among others. CRISPR-activation gene editing of *WRKY29* was carried out in *Solanum lycopersicum*. It was concluded that this TF intervenes in the homeostasis of Aux and CKs and the induction of somatic embryogenesis, in turn activating multiple other genes [78]. Similarly, crosstalk between *WRKY23* and auxin signaling pathways in *Arabidopsis* embryos was confirmed, which in turn regulates *WOX8* and *WOX9* [79]. In *Brassica napus*, *WRKY10* intervenes in the CK degradation pathway by regulating *CKX2* [80]. In *Solanum chacoense*, *WRKY* were shown to play a specific role in embryogenesis by detecting a high transient expression in torpedo embryos [81]. In *Gossypium hirsutum*, *WRKY14* and *WRKY40* were expressed only in embryos and not in the rest of the tissues studied, in addition to being related to stress response [82]. In *WRKY2* mutants in *Arabidopsis*, cell division does not occur normally, which also affects embryo formation [83].

The second family of transcription factors with the most differentially expressed members was *NAC*, comprising 38 genes, of which we identified a group negatively expressed in all stages of development, consisting of *NAC075a*, *NAC022c*, *NAC058a*, *NAC058c*, *NAC038c,* and *NAC067*. The rest of the *NACs* had very varied expression patterns, although upregulation predominated in the late stage. *NAC053* and 47a were the most expressed at sizes 7 and 8 cm, respectively. *NAC* TFs, which stand for NAM, ATF, and CUC, play a crucial role during embryogenesis. They have been associated with embryo patterning, cotyledon formation and division, establishment of the apical meristem, and the connection between maternal tissue and the embryo [84]. A member of this family, *NAC19*, was a key regulator in *Arabidopsis* embryogenesis, causing the embryos in overexpressed mutants to develop faster than in the wildtype, the seed size was larger, and it caused an increase in *YUC1* expression [85]. In contrast, NAC gene knockout mutants exhibited abnormal embryos, and their development was compromised from the torpedo stage onwards [86].

Although *AGL* is a large family, few of its members have been studied in detail for their involvement in embryogenesis. Here, most of the 17 identified *AG* and *AGL* family members were upregulated at nearly all fruit sizes. In contrast, *AGL6a* and *AGL7* were only locally expressed at sizes 2 and 3 cm, respectively. *AGL61* and *AGL65* were downregulated from 7 to 9 cm, while *AGL6b* was strongly repressed at a size of 5 cm. In *Arabidopsis*, *AGL15* and *AGL18* have been confirmed to be key regulators of somatic embryogenesis and to have redundant functions [87]. In mutants with overexpression of *AGL15*, embryogenesis occurred more rapidly, also proving that *AGL* in turn regulates other key genes such as *ABSCISIC ACID-INSENSITIVE3* and *FUSCA3* [88].

The *AP2/ERF* superfamily is one of the most numerous in plants and is related to embryogenesis; however, little is known about its mechanism of action [89]. Five of the six *AP2/B3-like* genes analyzed were positively expressed at the 3 cm size, then repressed and reactivated again starting at the 7 cm size. Only copy *c* showed negative expression values at the 8 cm size. This family mediates the transcriptional response through the effect of CKs and regulates embryogenic development in *Arabidopsis* [90] and *Z. mays* [91]. In *Hevea brasiliensis*, a specific pattern of expression of this superfamily was found during all stages of somatic embryos [92], consistent with our findings, where the expression of the various members is upregulated at all stages of embryogenic development. In the *ERF* family, we did not detect an expression pattern as evident as in the previous genes. *ERF9* was highly expressed only at the 2 cm fruit size, whereas *ERF11* and *ERF59c* were highly expressed at the 9 cm size. *ERF59b* remained highly expressed from the 7 cm size onwards, whereas the *ERF59a* copy was downregulated at the 2 and 5 cm sizes and then upregulated at the 7 cm size. In *Medicago truncatula*, *ERF*, induced by ethylene, had a high expression during the development of globular somatic embryos [93]. The only member of the *RAP2* family that showed positive expression was *RAP2.11a* at a fruit size of 3 cm. The remaining *RAP2.11b* and *RAP2.6* were found to be repressed from fruit sizes 7 and 8, respectively. These results are consistent with what was seen in *Brassica rapa*, where *AP2* genes, particularly *RAP2*, had a higher expression in heart stage embryos compared to globular ones, thus controlling the beginning of embryogenesis, to subsequently having a decrease in the following stages [94].

This family of TFs is vital for the initiation of embryogenesis, although the exact mechanism remains under study [95]. Of the eight *WOX* factor members, four were downregulated in at least one of the fruit sizes. The remaining four were upregulated in the late stage, with *WOX1a* being the most highly expressed, continuously from 7 cm to the end. *WOX1* is involved in meristem formation [96]. *WOX2* and *WOX8* are required for proper proembryo formation; furthermore, these genes crosstalk with the auxin efflux carrier *PIN1* [97]. In addition to *WOX8*, *WOX9* plays a redundant role in proper suspensor development [98].

Additionally, we searched for DEGs that have been reported to play an important role in embryogenesis (Appendix A), which were not included in the previous classifications. The most abundant family was the *Late Embryogenesis Abundant* (*LEA*) with 16 genes, most of which were found to be downregulated at almost all stages; except for copies d-f and m, whose expression was positive at the 7 cm size. As their name suggests, these proteins were discovered in the final stage of embryo development in *G. hirsutum* [99]. They are important towards the end of seed development, as they facilitate their tolerance to desiccation [100].

SCF^TIR1/AFB^ is a complex of proteins (SKP1, cullin, and F-box protein) whose function is the perception of Aux. Of these, the F-box proteins are the ones that confer the specificity to the complex by participating in the reception of Aux [101,102], and S-phase kinase-related protein 1 (SKP1) serves as a binding bridge between cullin and F-box proteins [103]. There are more than 700 F-box proteins identified in the *Arabidopsis* genome [104], whose functions are involved in various plant processes, such as embryogenesis and response to pathogens, among others. We identified members of the F-box family, comprising 14 isoforms, which are predominantly expressed at a low level during the late stage of development. However, we cannot assume that all these isoforms are participating in the embryogenic process. For its part, *SKP1-like 3* was detected early and late in development with negative expression, and *SKP1-like 4* was upregulated in late development with a pattern similar to *SLEEPY2* (*SLY2*), which is an F-box protein involved in gibberellin signaling. Loss-of-function mutants cause dwarf phenotypes, infertility, and seed dormancy [105].

In smaller numbers, we identified genes from the *Calreticulin* (*CRT*), *Maternal Effect Embryo Arrest* (*MEE*), *Protein phosphatase 2A* (*PP2A*), and *Somatic Embryogenesis Receptor Kinase 1* (*SERK1*). CRT is involved in cell division, as it has been found to accumulate highly, mainly in growth and regeneration zones, such as apical and root meristems. Furthermore, there are reports of increased expression in early stages of both zygotic and somatic embryos in models such as maize, *Daucus carota*, *N. plumbaginifolia*, and *C. canephora* [106,107,108]. Here, all five *CRT* isoforms were found to be upregulated at all sizes, except for 2 and 4 cm; *CRT2* was the most highly expressed, detected in the later stages of development. The enzyme adenylate cyclase (AC) catalyzes the synthesis of 3′,5′-cyclic adenosine monophosphate (cAMP), which acts as a second messenger regulating various plant processes such as embryogenesis [109]. In *Arabidopsis*, few proteins with AC activity have been identified, including an MEE protein whose function has already been confirmed [109]. *MEE* is expressed in correlation with other genes involved in embryogenesis and, in turn, regulates Aux biosynthesis genes such as *YUC4*, thus impacting embryogenic development [109]. *MEE66* was the only gene with positive expression in the 5–8 cm range, while *MEE60* and *MEE9* maintained negative values. Although *SERK1* plays a fundamental role in somatic embryogenesis, it has also been identified in zygotic embryogenesis. In *Arabidopsis*, it was expressed in early stages of zygotic embryos, specifically in those areas where a transition between developmental stages takes place [110]; a pattern similar to ours, where *SERK1* was upregulated at the 3 cm size. *PP2A* and *PP2A-3* exhibited specific expression in the 3 and 7 cm sizes, respectively. There is strong evidence that *PP2A* is essential for normal embryonic development. Loss-of-function mutants produced aberrant embryos, and auxin transport was also compromised [111].

Finally, the most prominent family, with 67 members, was *Uridine diphosphate (UDP)-glycosyltransferases* (*UGTs*), most of which were highly expressed starting at 7 cm. Although their direct relationship with embryogenesis has not been elucidated, 149 *UGTs* were identified in *Glycine max*, of which the maximum expression was located in globular embryos and early stages of seed maturation [112]. In addition, *UGTs* have been related to the conjugation of IBA [113] and to the control of IAA [114] and CK levels [115].

In summary, the genes with the highest levels of expression in the final stage of development include *UGT71C4b*, *CRT2*, and *SLY2*.

The results of differential expression analysis can be influenced by both the sample used as a control and the fold change thresholds and *p*-values used to define differential expression. To complement this approach and reduce potential biases, gene co-expression network analysis was performed using WGCNA in this work. For this purpose, the 50% most variable genes were used, which allowed excluding those with null expression values in all samples. The data were transformed to log_2_(TPM+1) and a soft threshold value of 20 was applied, determined from the scale independence and mean connectivity plots shown in Appendix A, along with the clustering dendrogram.

As a result of this analysis, 22 gene co-expression modules were identified. No module showed a strong correlation (>0.85) with any developmental stage of the avocado zygotic embryo, as shown in Appendix A. However, key transcription factors (TFs) of zygotic embryogenesis were identified in the blue module, which exhibits a high expression pattern from 1 cm to 5 cm that decreases from 7 cm onwards, as shown in Figure 7A. Among the identified TFs are *WOX1*, *WOX2*, *WOX13*, *LEC1* (*NFYB6*), *CUC2* (*NAC098*), *CUC3* (*NAC031*), and *FUS3*. In *A. thaliana*, *WOX1* and *WOX2* are needed for the establishment of the SAM, while *LEC1-2* and *FUS3* are key genes regulating seed maturation [116]. Additionally, CK metabolism genes were identified such as *LOG8*, *CKX1*, *CKX5*, *ARR1*, *AAR3*, *ARR9*, *ARR11*, *CRF4*, as well as *AHK1* and *AHK5*. This module was also found to be enriched with genes that regulate cell division, including *CYCD3-1* and other *CYCD* genes, whose expression increases in response to CKs and plays an important role in cell division during seed development [117]. Finally, within this module, indole acetic acid (IAA) metabolism genes were identified, such as *YUC4*, *PIN1*, *ARF5*, *ARF6*, and *ARF19*. The interaction network of genes belonging to this module is presented in Figure 7A. This network is divided into three functional groups: the key TFs are found within the auxin metabolism group. In contrast, genes involved in cell division connect with those of CK metabolism through *CYCD3-1*, which acts as a connector between the two functional groups.

In contrast to the blue module, the red module shows high expression from 7 cm onwards, as observed in Figure 7B. In this module, genes mainly involved in starch metabolism, and abscisic acid biosynthesis, degradation, and signaling pathways were mostly identified, among which are *ZEAXANTHIN EPOXIDASE* (*ZEP*), *XANTHOXIN DEHYDROGENASE* (*ABA2*), *9-CIS-EPOXYCAROTENOID DIOXYGENASE* (*NCED3*), *ABSCISIC ACID RECEPTOR* (*PYR1*), *ABSCISIC ACID RECEPTOR* (*PYL9*), *ABSCISIC ACID 8′-HYDROXYLASE 2* (*CYP707A2*) and *ABSCISIC ACID-INSENSITIVE 5* (*ABI5*) [118]. Additionally, in this module, there are also water deprivation response genes that could respond to water loss during the late stage of seed development [119]. Genes found in the blue module play key roles during seed maturation, while genes identified in the red module are associated with the acquisition of tolerance to desiccation and dormancy [120].

Clearly, Aux and CKs play a crucial role in the formation and development of the zygotic embryo. Our results suggest that there is active Aux synthesis and transport (Figure 8), and at the same time, strict regulation of its signaling pathway. The same is true for CKs’ homeostasis. There is de novo transcription of genes involved in CK biosynthesis, as well as of phosphorylases involved in its signaling pathway (Figure 8). The data on the expression of genes involved in Aux and CK homeostasis obtained in this study allow us to propose the model presented in Figure 8. The endogenous concentration of Aux and CK pools allows the embryo to develop until it establishes its position within the fruit.

## 3. Materials and Methods

### 3.1. Biological Material and RNA-Seq

*P. americana* fruits of different sizes were collected in Michoacán, Mexico, from which zygotic embryos (ZE) were extracted: 1, 2, 3, 4, 5, 7, 8, and 9 cm, with three replicates per size. Around 0.1 g of ZE was used for RNA extraction [121]. The libraries were generated by Novogene (Sacramento, CA 95817, USA) in paired-end mode using the Illumina platform, with a depth of 40 million reads and a read length of 150 bp per sample. Raw reads were deposited in the NCBI Sequence Read Archive (SRA) database with the accession ID PRJNA1297176.

### 3.2. Bioinformatic Analysis

Default values were used in all software unless otherwise stated. Raw reads were subjected to quality analysis using FastQC v0.12.1. Subsequently, alignment to the reference genome [122] was performed using HISAT v2.1.0 [123]. Gene counts were obtained with HTSeq v2.0.5 [124]. Differential expression analysis was performed with DESeq2 (v1.22.1) [125] using ZE 1 cm as a reference for comparisons, *P-*adjusted value < 0.05, and log2FC 1.5 (<1.5 for downregulated and >1.5 for upregulated genes).

For Gene Ontology (GO) analysis, the corresponding *A. thaliana* orthologs of the upregulated genes were entered into the DAVID tool [126]. All non-redundant *A. thaliana* orthologs of DEGs were used as background, and a *P*-value cutoff of 0.05 was applied. Subsequently, the five terms with the lowest *p*-value from each category were selected and plotted using ggplot2 v3.5.1 in RStudio 2024.04.1+748.

The functional annotation of the *P. americana* genes was performed by selecting the best hit of homologous genes from *A. thaliana* with DIAMOND v0.9.25 [127] using 1 × 10^−6^ as e-value threshold. Representative model proteins of *A. thaliana* were downloaded from the TAIR portal (accessed 8 April 2024). TFs were identified using the online tool PlantTFDB [128] (http://planttfdb.gao-lab.org/; accessed on 22 May 2025). To construct the co-expression networks, expression values were transformed using log2(TPM+1), and the 50% most variable genes (based on standard deviation) were selected for further analysis. To obtain the co-expression modules, WGCNA v1.72-5 [129] was used with the following parameters: soft_power = 20 (chosen according to the scale independence and mean connectivity graphs), TOMType = “signed”, networkType = “signed”, minModuleSize = 50, mergeCutHeight = 0.25, deepSplit = 2, randomSeed = 123, corType = “pearson”). The modules were correlated with the seed sizes. The “moduleTraitCor” and “moduleTrait*P*value” functions from the WGCNA library were applied, which calculated and evaluated the statistical significance of the correlations between gene modules and their sizes.

## 4. Conclusions

Genes related to auxins, cytokinins, transcription factors, and others that were actively involved during the different stages of avocado fruit development were identified. A clear pattern of gene expression was observed for these genes, with a noticeable difference between the initial and intermediate stages, compared to the late stages. The number of differentially expressed genes increased towards the later stages, with several important families standing out, including *ABC*, *PIN*, *YUC*, *GH3*, *ARR*, *AGL*, *WRKY*, *WOX*, *LEA*, and *UGT*, among others (Figure 9).

We can speculate that in the early stages, cell growth and division occur, and the apical and basal axes are established, primarily identifying genes involved in auxin transport. Later, the embryo begins to differentiate and accumulates the necessary metabolites and reserves for its growth. When the final stages of development are reached, stress tolerance mechanisms are activated as the embryo prepares for the period of desiccation and dormancy.

Transcriptome analysis of avocado ZE will serve as a basis for studying the molecular mechanisms that govern this process, thereby facilitating the exploration of various biological questions in future studies. The results obtained in this analysis will also aid in understanding and transferring what occurs during somatic embryogenesis, enabling the establishment and optimization of this process, which would be particularly useful since avocado is a recalcitrant species of agricultural interest.

## Figures and Tables

**Figure 1 plants-14-03288-f001:**
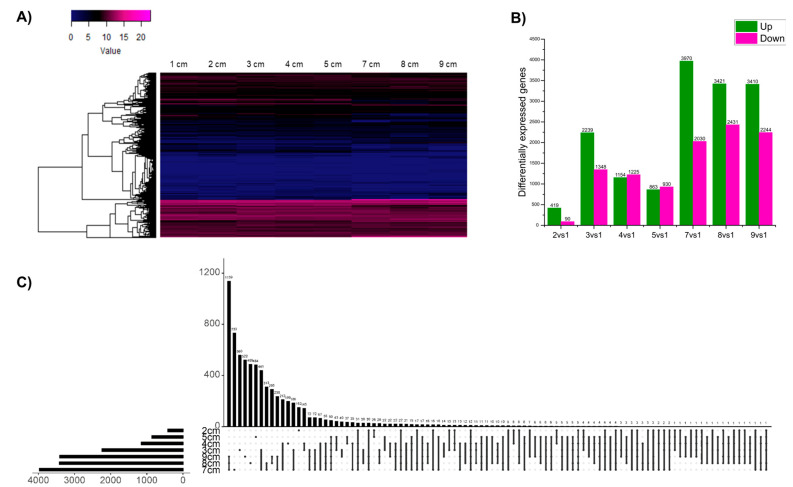
**Transcriptomic analysis of different ZE stages in *****P. americana***. (**A**) Heatmap showing the global expression profile. (**B**) Differentially expressed genes number per stage, using 1 cm fruit size as reference (log2FC > 1.5 and *P*-adjusted < 0.05). (**C**) UpsetR plot of the upregulated DEGs. The overlapping regions correspond to the number of shared genes between conditions.

**Figure 2 plants-14-03288-f002:**
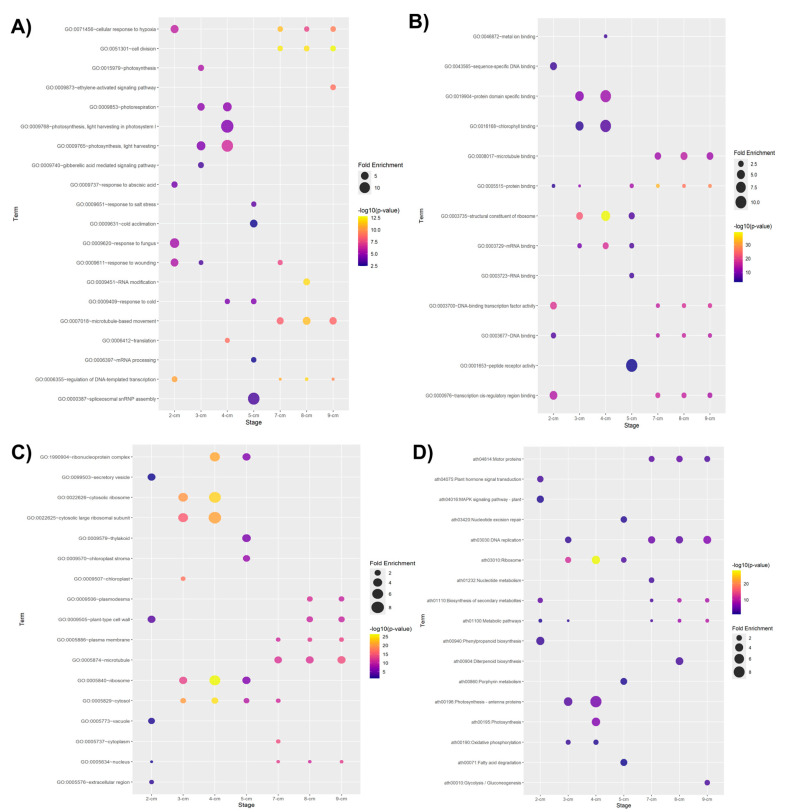
**Gene ontology enrichment analysis of upregulated DEGs.** Gene ontology (GO) enrichment belonging to (**A**) biological process, (**B**) molecular function, and (**C**) cellular component. (**D**) KEGG pathways enrichment. The corresponding *A. thaliana* orthologs of the upregulated genes were entered into the DAVID tool.

**Figure 3 plants-14-03288-f003:**
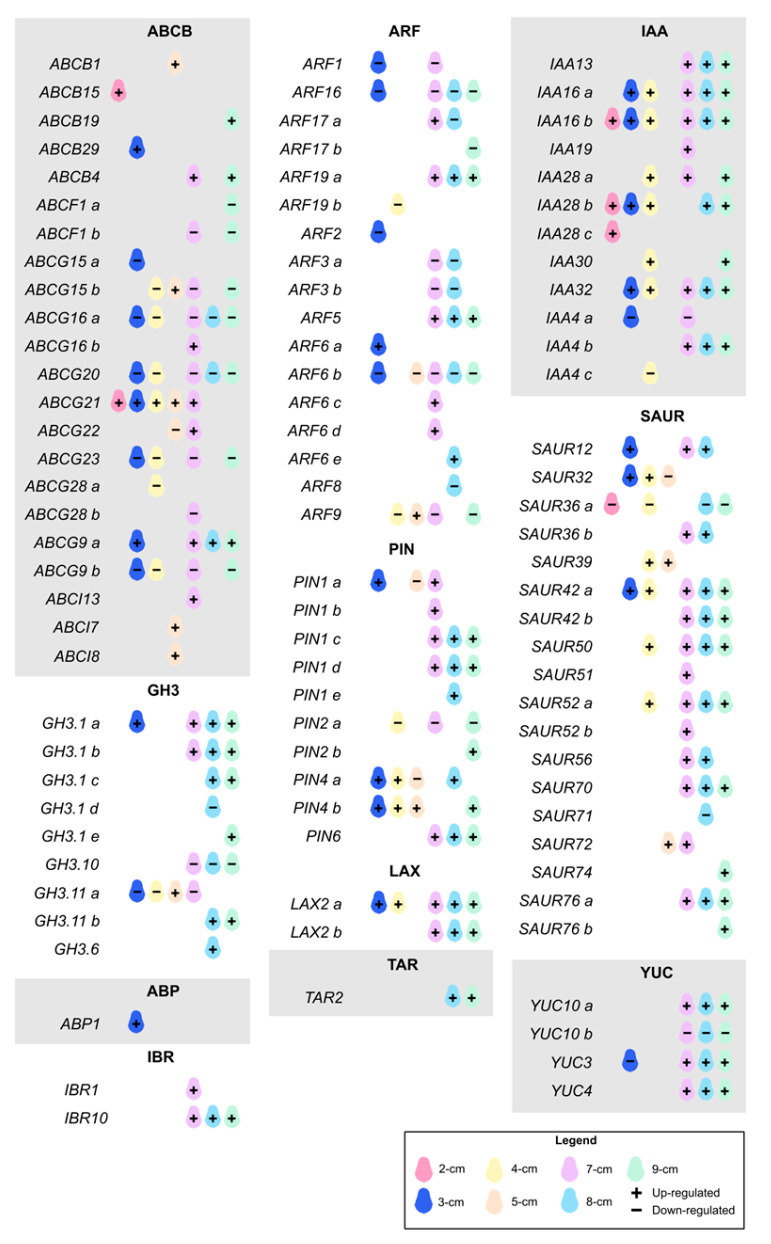
**Expression patterns of auxin-related genes during the different ZE stages in** ***P. americana.*** This diagram was created manually using Krita v5.2.9 software, with data corresponding to the differential expression of auxin-related genes. Fruit sizes are shown in different colors. The + and − signs indicate that the gene is up- or downregulated under each specific condition. These data are also represented in a heatmap in Appendix A.

**Figure 4 plants-14-03288-f004:**
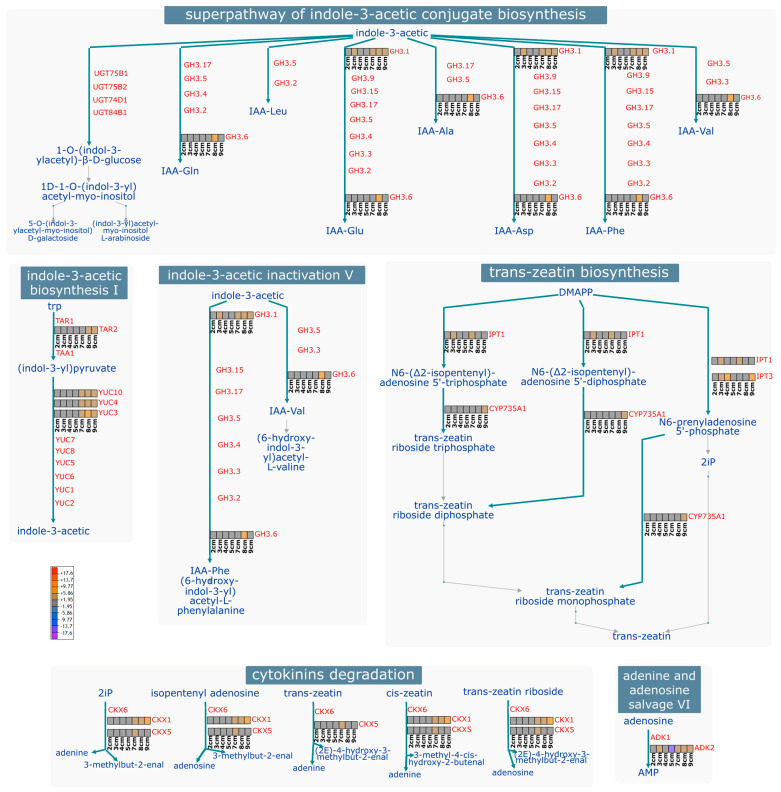
**Metabolic map of some enriched DEGs.** This diagram shows the most enriched pathways involving some of the DEGs. Differential expression data were incorporated into the Plantcyc platform using *A. thaliana* as the background for developing the model.

**Figure 5 plants-14-03288-f005:**
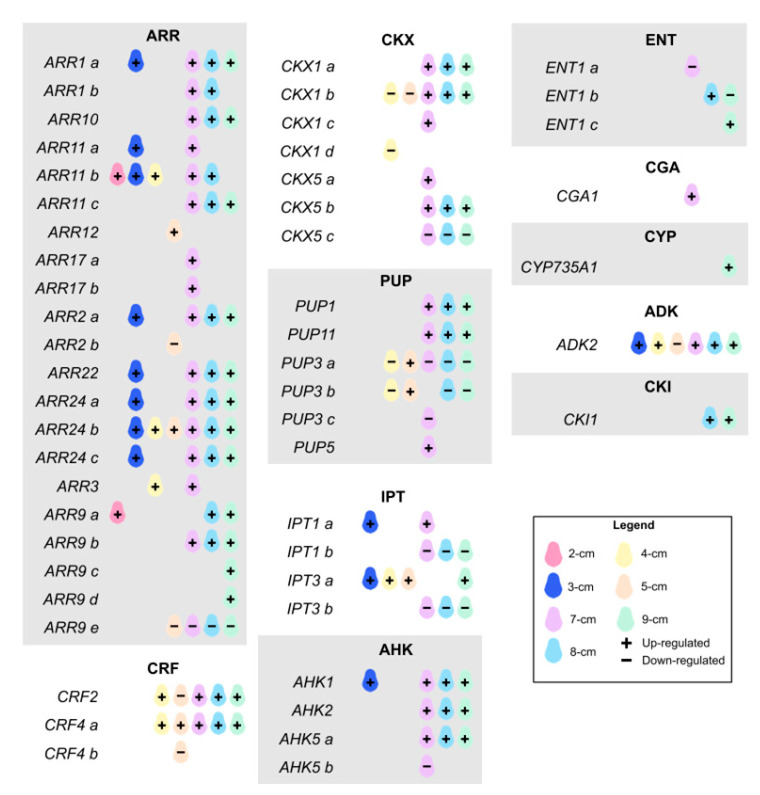
**Expression pattern of cytokinin-related genes during the different ZE stages in** ***P. americana*****.** This diagram was created manually using Krita v5.2.9 software, with data corresponding to the differential expression of cytokinin-related genes. Fruit sizes are shown in different colors. The + and − signs indicate that the gene is up- or downregulated under each specific condition. These data are also represented in a heatmap in Appendix A.

**Figure 6 plants-14-03288-f006:**
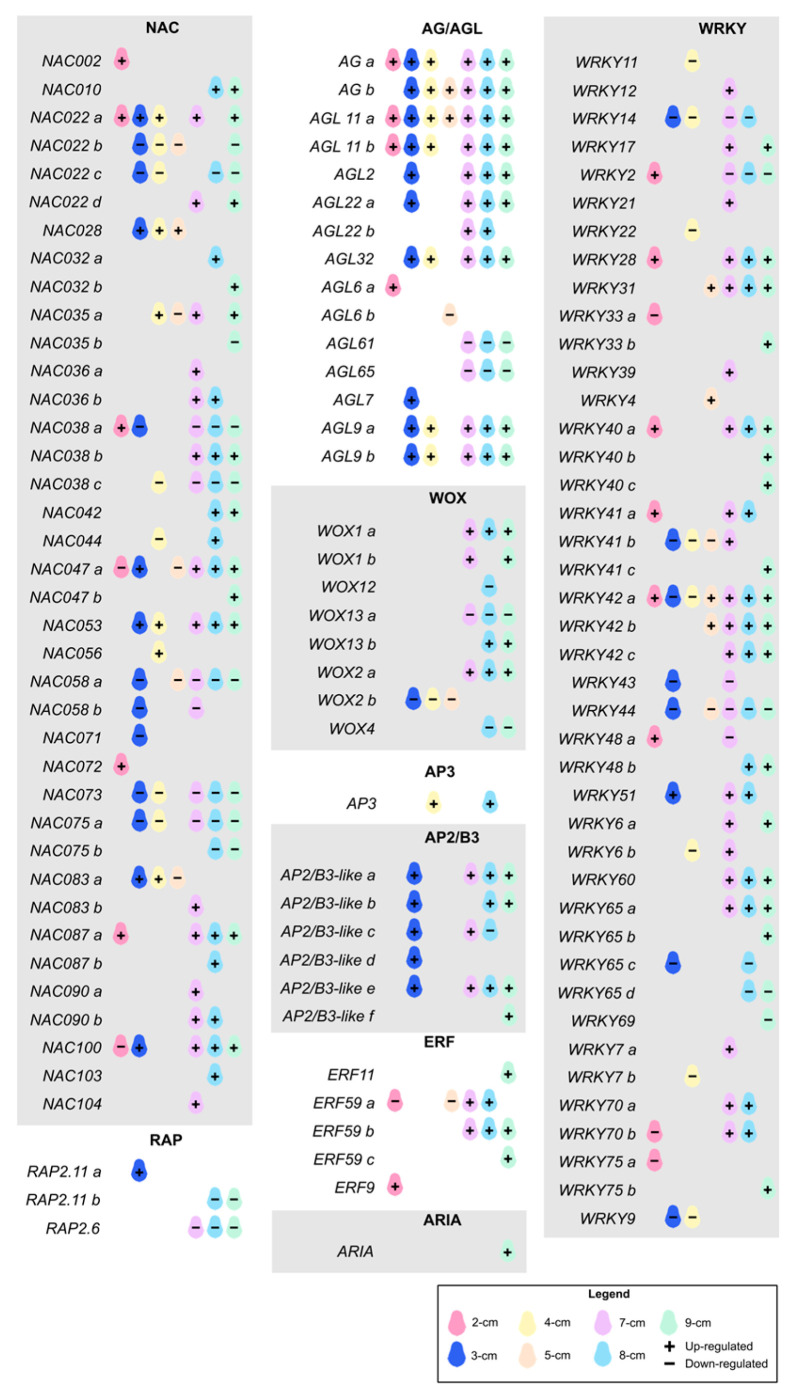
**Expression pattern of transcription factors during the different ZE stages in** ***P. americana*****.** This diagram was created manually using Krita v5.2.9 software, with data corresponding to the differential expression of transcription factors. Fruit sizes are shown in different colors. The + and − signs indicate that the gene is up- or downregulated under each specific condition. These data are also represented in a heatmap in Appendix A.

**Figure 7 plants-14-03288-f007:**
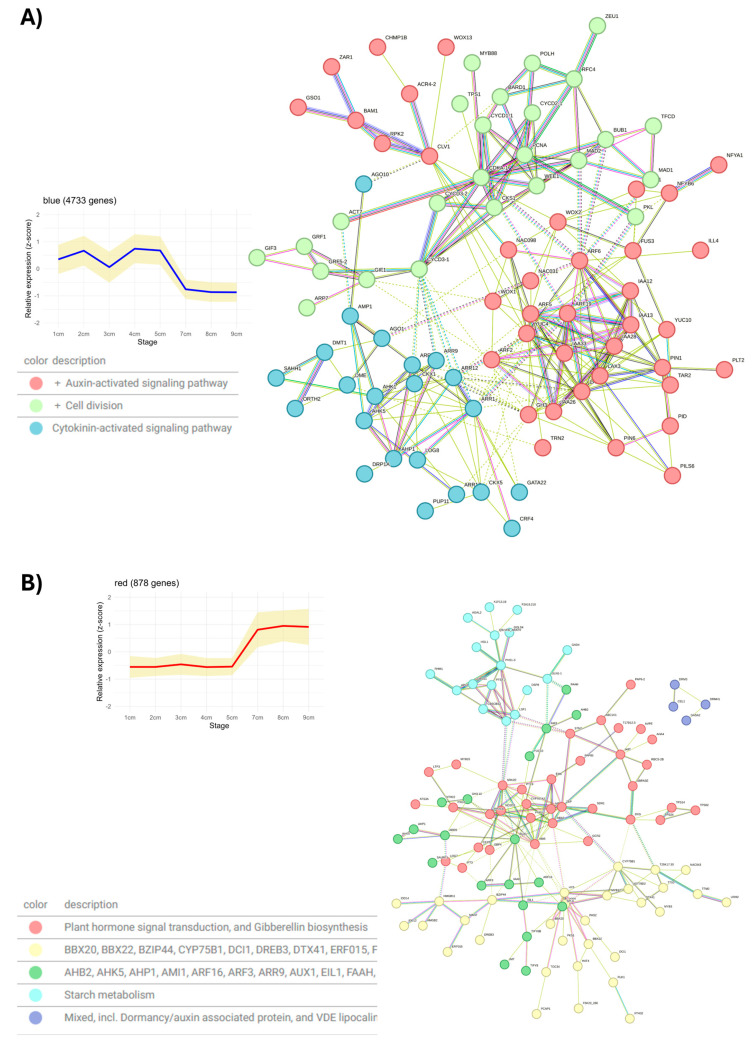
**Functional analysis of** ***P. americana*** **gene modules containing key genes for seed development.** (**A**) Blue module. (**B**) Red module. The average expression profile of each co-expression module, along with the functional association network constructed using the STRING database, is shown. Edges represent protein–protein associations. The grouping of the clusters is observed in different colors.

**Figure 8 plants-14-03288-f008:**
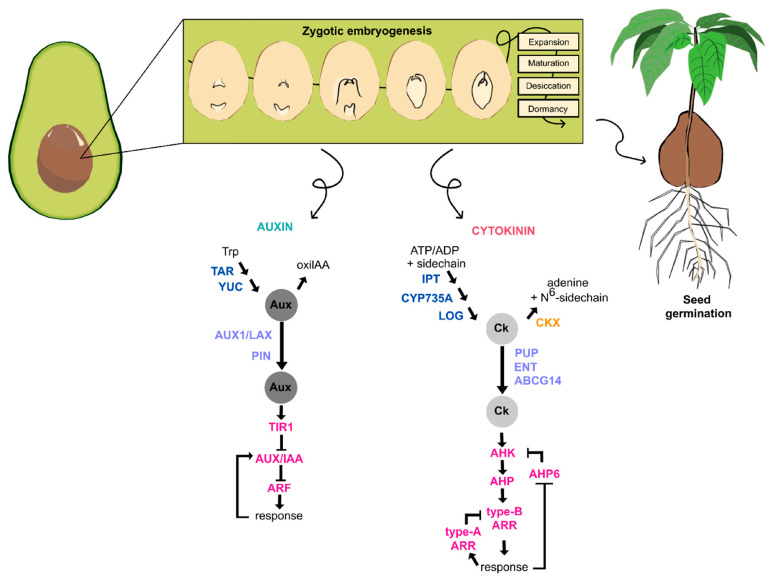
**Proposal for the participation of auxins and cytokinins during zygotic embryogenesis of** ***P. americana***. Schematic representation summarizing the different stages of development of the zygotic embryo within the seed and the main pathways and genes involved in the homeostasis of the auxin and cytokinins during the process. The embryos are not drawn to scale.

**Figure 9 plants-14-03288-f009:**
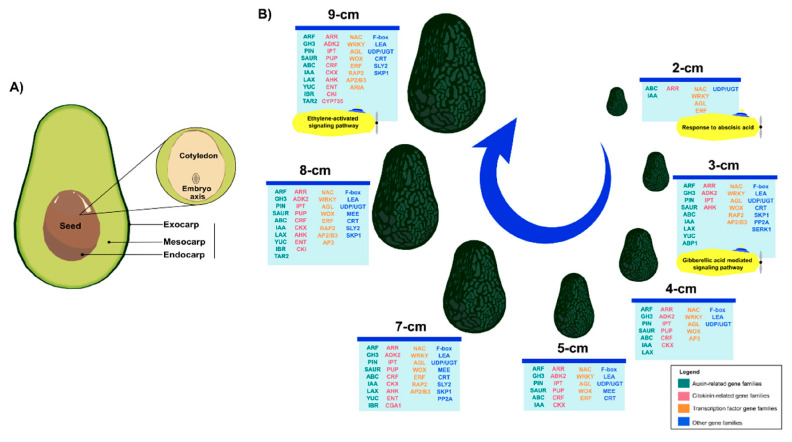
**Exploring the seven stages according to fruit sizes from** ***P. americana*****.** (**A**) Schematic representation of avocado fruit and seed cross-section. (**B**) Principal gene families involved during embryo development from different avocado fruit sizes.

## Data Availability

Raw reads were deposited in the NCBI Sequence Read Archive (SRA) database with the accession ID PRJNA1297176.

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
