# Peer review of "Identification of Auxin, Cytokinin, Transcription Factors, and Other Zygotic Embryogenesis-Related Genes in *Persea americana*: A Transcriptomic-Based Study [Author-notes fn1-plants-14-03288]"

_plants, 2025, doi:10.3390/plants14213288_

Round 1

Reviewer 1 Report

Comments and Suggestions for Authors

This paper, although not having novelty in the subject or in the techniques used, is of very high interest for scientists doing research in tropical fruits, particularly with acocado. Also, the reported results and discussion are of much interest in understanding the somatic embryogenesis, especially in woody species. The writing would benefit of a new readind to eliminate redundancy and improve.

Author Response

Dear Editor,

I have attached the corrected manuscript “Identification of auxin, cytokinin, transcription factors, and other zygotic embryogenesis-related genes in Persea americana: a transcriptomic-based study” (Manuscript ID: plants-3883249).

We have carefully followed the three reviewers' recommendations. We have eliminated redundancies and citations, as they did not recommend.

Additionally, we carefully reviewed the manuscript, correcting a couple of typos and ensuring the names of all genes were in italics, as some were not.

We replaced Figure 2 with a higher-quality version, as the previous one did not meet the required standards.

We include both versions, one with change control and one without.

We thank the reviewers for their work and hope that this version is suitable for publication.

Reviewer 2 Report

Comments and Suggestions for Authors

The authors have presented a clear and well-structured manuscript. The work is valuable and will serve as an excellent starting point for researchers specializing in avocado and zygotic embryogenesis. I found the work to be of high quality and have no additional comments or suggested revisions at this time

Author Response

(The authors gave the same response as above.)

Reviewer 3 Report

Comments and Suggestions for Authors

As far as my competence allows me to judge, it is a well-written and scientifically sound article which presents significant results. I find this manuscript to be suitable for publication in present form and I do not have any reasonable advices for improvement.

Author Response

(The authors gave the same response as above.)
